# Testing and Validation of a Custom Retrained Large Language Model for the Supportive Care of HN Patients with External Knowledge Base

**DOI:** 10.3390/cancers16132311

**Published:** 2024-06-24

**Authors:** Libing Zhu, Yi Rong, Lisa A. McGee, Jean-Claude M. Rwigema, Samir H. Patel

**Affiliations:** Department of Radiation Oncology, Mayo Clinic, Phoenix, AZ 85054, USA; zhu.libing@mayo.edu (L.Z.); mcgee.lisa@mayo.edu (L.A.M.); rwigema.jean@mayo.edu (J.-C.M.R.)

**Keywords:** LLM, HN cancer, supportive care, symptom management

## Abstract

**Simple Summary:**

Cancer patients, especially long-distance patients, often struggle to receive timely and precise medical information and support for their symptom management and survivorship care. ChatGPT-4’s responses to queries concerning head and neck (HN) cancer remain questionable. The purpose of this study was to develop and validate a retrained large language model (LLM) for HN cancer patients. In this cross-sectional study, the presented LLM was retrained with a high-quality user-defined knowledge base. The responses from the LLM to patients’ questions were validated against human responses, and the model showed a superior performance, with average scores of 4.25 for accuracy, 4.35 for clarity, 4.22 for completeness, and 4.32 for relevance, on a 5-point scale. The confined-trained LLM with a high-quality user-defined knowledge base demonstrates high accuracy, clarity, completeness, and relevance in offering evidence-based information and guidance on the symptom management and survivorship care for head and neck cancer patients.

**Abstract:**

Purpose: This study aimed to develop a retrained large language model (LLM) tailored to the needs of HN cancer patients treated with radiotherapy, with emphasis on symptom management and survivorship care. Methods: A comprehensive external database was curated for training ChatGPT-4, integrating expert-identified consensus guidelines on supportive care for HN patients and correspondences from physicians and nurses within our institution’s electronic medical records for 90 HN patients. The performance of our model was evaluated using 20 patient post-treatment inquiries that were then assessed by three Board certified radiation oncologists (RadOncs). The rating of the model was assessed on a scale of 1 (strongly disagree) to 5 (strongly agree) based on accuracy, clarity of response, completeness s, and relevance. Results: The average scores for the 20 tested questions were 4.25 for accuracy, 4.35 for clarity, 4.22 for completeness, and 4.32 for relevance, on a 5-point scale. Overall, 91.67% (220 out of 240) of assessments received scores of 3 or higher, and 83.33% (200 out of 240) received scores of 4 or higher. Conclusion: The custom-trained model demonstrates high accuracy in providing support to HN patients offering evidence-based information and guidance on their symptom management and survivorship care.

## 1. Introduction 

Cancer patients encounter significant challenges in obtaining pertinent and trustworthy medical information on symptom management and survivorship care [1,2]. As outpatient numbers grow and the need for efficient care increases, these patients often struggle to obtain straightforward advice on how to handle side effects, maintain their mental health, and manage long-term care [3]. It can be quite disruptive and debilitating to not have intermediary access and guidance from professionals on managing nausea and related health symptoms post radiotherapy [4]. This predicament is particularly acute for patients residing at a distance far from medical care, who may feel less equipped to manage side effects at home [5]. While there exist standardized guidelines from reputable sources for post-treatment management and supportive care, they remain scattered and heterogeneous in their present forms. Consolidating such information from disparate sources is often difficult and time consuming.

Large language models (LLMs) have exhibited remarkable proficiency in processing and understanding natural language and generating responses across different medical domains. Introduced in 2018 by OpenAI, ChatGPT, a generative pretrained transformer (GPT), is widely now known for its capabilities to provide human-like responses by collating information from various publicly available sources. Such a GPT has started to highlight substantial medical knowledge [6] and an ability of medical reasoning [7,8]. It was reported that GPT-4 can achieve notable and passing scores on Steps 1–3 of the United States Medical Licensing Examination (USMLE) over a 20-point margin, making it the first computer-based system to qualify for a standardized licensure examination [9]. Remarkably, a 540-billion parameter LLM [10], the Pathways Language Model, achieved state-of-the-art scores on the USMLE with 67.6% accuracy [7]. It is now broadly believed that further fine-tuning LLMs with additional medical licensing questions and answers selected by clinical experts can further improve the responses required by an LLM for medical reasoning. As is also the case in our study, previous reports have shown ChatGPT’s outstanding performance on suggesting an initial diagnosis, examination steps, and treatment across various clinical disciplines [11,12].

Despite its potential to provide medical information, GPT-4’s training data are not based on specialized medical knowledge [13,14,15] as it gathers most of its information from the public domain [16], of which its validity sometimes remains questionable. OpenAI could be liable for medical misinformation [17]. The present state-of-the-art model raises concerns about potential medical misinformation for cancer patients who often seek timely health care advice and information online, thereby exposing themselves to the perils of inaccurate information [18,19]. Of paramount concern now is the wide access to GPT-4 platforms that, if not validated, can lead to a widespread exposure to such medical information from ChatGPT that may or may not be accurate [20,21,22,23,24]. A recent research study reported that ChatGPT may contain fundamentally flawed or inadequate information in the field of urology oncology due to the extrapolation of data from the relevant literature and abstracts without considering the logic or accuracy [25]. In a comparative study with the National Comprehensive Cancer Network (NCCN) guidelines, ChatGPT concurred only 61.9% of the time on treatment recommendations across 104 input prompts in five assessed criteria [18]. Moreover, as scientific research evolves rapidly and LLMs lack periodical updates, their access to the latest findings and novel guidelines could remain limited in the realms of care discussed [26]. Minor discrepancies were observed when querying different versions of GPT-3.5 [8], indicating that current LLM-based chatbots may not be sufficiently reliable enough for direct use by cancer patients.

In order to enhance the accuracy of such chatbots, several studies have focused on incorporating high-quality publications into chatbot frameworks for specific areas like urology oncology [25] and prostate cancer [27]. Integrating the European Association of Urology (EAU) oncology guidelines with GPT has shown promise in addressing questions that ChatGPT-4 previously struggled to answer accurately [25]. Furthermore, fine-tuning LLMs has now been conducted by others to tailor the models to specific medical inference tasks [28]. Specialized training and resources can promote broader applications in medical practice [29]. However, there is a paucity of data for developing and validating an LLM with user-defined knowledge for the supportive care of head and neck (HN) cancer patients.

The novelty of this study lies in folding both high-quality guidelines and responses from our institution’s physicians’ and nurses’ answers to the patient clinical questions into the model. This enables our application to provide medical information for HN cancer patients solely from our predefined knowledge base, thereby offering a secured guardrail. With this guardrail in place, we hypothesize that our retrained LLM can extract more accurate data without resorting to various online sources. Additionally, to lessen the burden on physicians reviewing the answers, we developed a method for automatically evaluating the chatbot’s performance.

## 2. Materials and Methods

### 2.1. High-Quality User-Defined Database

The high-quality training datasets for our model are composed of two key parts. Firstly, we compiled a dataset consisting of questions and answers sourced from patient portal messaging within Epic (Epic Systems Cooperation, Verona, WI, USA), an Electronic Medical Record (EMR) platform utilized by nurses and physicians to correspond with patients on their queries and clinical situations during and after the completion of their radiotherapy. We curated 150 questions and answers from 90 head and neck (HN) cancer patients. These were anonymized per our institutional protocol to remove patient, physician, and nurse identifiers. In this cohort, there were 8 nasopharynx, 28 oral cavity, 30 oropharynx, 13 larynx, and 11 tongue cancer cases treated between January 2021 and January 2023. These real patient inquiries predominantly focused on side effects experienced during and after radiotherapy and were addressed by nurses and physicians from our institution.

The second component of our dataset comprises authoritative guidelines on supportive care for HN cancer patients. These guidelines were published by reputable organizations such as the American Society of Clinical Oncology, the American Cancer Society, and the National Comprehensive Cancer Network [30,31,32,33,34,35,36]. They involve various aspects, including managing physical side effects, living as a cancer survivor with head and neck cancers, and general supportive care. These real patient inquiries, combined with established medical guidelines, form a comprehensive and dependable source of information for training our model, with the goal to furnish accurate and beneficial responses to HN cancer patients.

### 2.2. Workflow of Custom-Trained Chatbot

We propose a specialized LLM-based model retrained using a high-quality external database operating within a Gradio interface [37] that facilitates the training of ChatGPT-4 with the custom external knowledge base. Firstly, the temperature parameter that controls the randomness of text generation with GPT was set to 0.2. A lower temperature contributes to a more deterministic response. The nucleus sampling parameter was set to 1, allowing GPT to consider all possible input tokens. Additionally, the presence penalty was set to 1 to reduce repetitive tokens, and the frequency penalty was also set to 1 to penalize the generation of frequent tokens. Lastly, the maximum generation was set to 500 words to avoid overly lengthy responses. Such a constraint ensures that the LLM produces succinct and articulated response with less room for hallucination and derailments. The retrained LLM was implemented on an NVIDIA GeForce RTX 2080 Ti. The workflow of this application is depicted in Figure 1.

Essentially, the process begins with inputting a range of files, including those in PDF and Word formats, into the training matrix. Subsequently, the large amount of text is segmented into small fragments of information using a chunking technique [38]. Vector embeddings (numerical vectors) are then employed to numerically represent words, phrases, and sentences. This vector database serves as a repository for both structured and unstructured data, thereby ensuring faster retrievals.

With a new patient query requested, the LLM interprets and analyzes the query’s content by converting the text into numerical vectors. The LLM response is then generated based on a vector search by calculating the distance between vector embeddings in the vector space [39]. The information is then finally retrieved from the predefined external database.

### 2.3. Testing and Validation of Custom-Trained Chatbot

Three board-certified radiation oncologists (RadOncs) specializing in the treatment of HN malignancies evaluated the LLM-generated responses to 20 questions based on four criteria: accuracy, clarity, completeness, and relevance. Each criterion was scored based on a Likert scale that displayed the extent of agreement between the physician team and the LLM-generated responses, (1: strongly disagree; 2: somewhat disagree; 3: somewhat agree; 4: agree; 5 strongly agree). The questions were divided into two groups: (1) 10 questions and answers from Epic, not included in the training dataset; and (2) 10 questions created using GPT-4, focusing on common patient concerns like xerostomia, dysgeusia, dysphagia, fatigue, anxiety, lymphedema, dental issues, dehydration, mucositis, and nausea. A T-test was employed to examine potential subjective bias between different RadOncs (*p* < 0.05 was considered statistically significance). Separately, a set of 10 additional questions were used to cross-compare our model’s response against those provided by the healthcare professionals to establish preferred answers by RadOncs.

### 2.4. Auto-Evaluation of Chatbot Answers

In addition, using GPT-4, an automated scoring method was also developed to evaluate and score these answers. This method aims to mitigate human bias and also reduce, in the future, the workload of manual scoring. This auto-evaluation mimics how an expert could interpret and score the quality of retrained LLM answers based on GPT-4’s understanding of the answers. In considering the Epic answers as a benchmark, the scoring was conducted through a subjective qualitative assessment and comparison between the retrained LLM answers and Epic answers. As discussed previously, the same four criteria were utilized: accuracy, clarity, completeness, and relevance. During the GPT-4 subjective assessment, an implicit comparison is observed where certain keywords or phrases might stand out as particularly related or missing, which can thereby influence the scores. For instance, if the Epic reference answer mentions “referral to physical therapy” and the model response does not, this is then considered a missing element. However, the evaluation process does not solely rely on systematic keyword matching; it involves a more comprehensive analysis of the overall content.

To validate this automatic scoring against the scores from the RadOncs, we used a statistical test called the t-test (paired two sample for means). This statistical test aims to determine if there is a significant difference between the two scoring methods. A result with a *p* value below 0.05 indicates a statistically significant difference.

## 3. Results

### 3.1. Testing and Validation on Chatbot Performance

The evaluation scores provided by the RadOncs on the 20 AI-based responses are illustrated in Figure 2 with the 20 questions used and retrained LLM’s responses available in the Appendix A. The average scores were 4.25, 4.35, 4.22, and 4.32 for accuracy, clarity, completeness, and relevance, respectively. The 95% confidence interval (CI) ranged between [4.00, 4.50], [4.13, 4.57], [3.96, 4.47], and [4.07, 4.57] for these criteria. Approximately 91.67% (55 out of 60) of the answers scored 3 or higher for accuracy, and 51.67% (31 out of 60) achieved a perfect score in accuracy, indicating that the RadOncs strongly agreed with the AI-generated responses. Clarity received the highest scores within the agreement rates (scores of 4 or higher) with 83.33% for accuracy, 88.33% for clarity, 80% for completeness, and 81.67% for relevance. The median score of three RadOncs was 5 for all criteria with a small portion of outliers (less than 6) observed for each criterion. The top 25% of scores were mostly 5, while the bottom 25% were mostly 4 for accuracy, clarity, and relevance, with the lower quartile for completeness being 3. The majority of scores for all four criteria fell between 4 and 5. The average scores for the 10 real patient questions were 3.90, 4.00, 3.85, and 3.95 for accuracy, clarity, completeness, and relevance, respectively. However, the scores for the 10 AI-generated questions were higher: 4.50 for accuracy, clarity, and completeness and 4.00 for relevance. This suggests that real patient questions, which often cover multiple issues, are more complex and harder to answer accurately.

Subjective bias was observed in the RadOnc evaluations, with differences between the highest and lowest scores ranging between 13.75% and 23.06% for each criterion. Notably, RadOnc 2 and RadOnc 3 showed significant differences in their scores for clarity (t = 2.0930; *p* < 0.05). No significant difference was found between RadOnc 1 and 3 for accuracy and clarity. For completeness and relevance, there were noticeable differences between both RadOncs 1 and 3 and RadOncs 2 and 3. Most of the relevance scores were 5 for RadOnc 3.

When comparing the answers from Epic with those generated by the retrained LLM, the three RadOncs showed different preferences. Six out of ten LLM answers were preferred by RadOncs 1 and 3, while RadOnc 2 preferred three of the LLM answers. They unanimously agreed on their preference for the LLM answers to three questions (Q2–4), suggesting that the retrained LLM may offer potentially better answers than human responses for head and neck cancer patient queries. Nonetheless, there is a noticeable difference in preferences among the different RadOncs, indicating the presence of some subjective bias.

### 3.2. Comparison of Auto-Evaluation and RadOnc Scores

The responses generated by the model for 10 Epic-based questions were evaluated by GPT-4 using the answers from nurses or physicians as a benchmark for reference. The scores are depicted in Figure 3a. The average scores were 4.2 for accuracy, 4.8 for clarity, 3.9 for completeness, and 4.4 for relevance. The distribution of these scores for accuracy and relevance provided by GPT-4 closely mirrored those given by the RadOncs; however, the RadOncs had more variation and outliers in clarity and completeness.

Figure 3b illustrates the difference in scores between the RadOncs and GPT-4. The smallest difference was 0.23 for relevance. This small difference is due to the low “temperature” setting of 0.2 in the LLM, resulting in responses closely related to the questions. Both the RadOncs and the automatic evaluation yielded high scores in this area. The difference in the accuracy score was just 0.2, indicating that our automatic method can evaluate almost as effectively as the RadOncs. The average score difference for completeness was −0.1. The RadOncs scored this higher, likely leveraging their expert knowledge, while GPT-4 scored this lower because it always compared to the Epic answer (the reference answer). However, the RadOncs might not be satisfied with the Epic answers potentially scoring higher in this category. The biggest difference was −0.67 for clarity, suggesting that GPT-4 tends to give lower scores than the RadOncs in this area.

The *p*-values between mean scores of the automatic evaluation and the three RadOncs were 0.40, 0.07, 0.17, and 0.50 for the four criteria. There was no significant difference between the GPT-4 and RadOnc mean scores.

## 4. Discussion

We developed a retrained LLM that integrates responses from both nurses and physicians along with high-quality guidelines for managing the side effects and supportive care of HN cancer patients undergoing radiotherapy. The LLM’s performance was validated by RadOncs across four criteria: accuracy, clarity, completeness, and relevance. Similar evaluation metrics have been utilized in previous studies [23,40,41]. However, our model evaluations did not include the empathy metric and patient questionnaire responses [27]. The mean scores of the accuracy and clarity were 4.25 and 4.35, respectively. Our retrained LLM (GPT-4) achieved higher scores in accuracy and clarity compared to previously published data for which GPT-3 was used for breast tumor question assessments. In those studies, the evaluators agreed or strongly agreed with the responses of the GPT-3 model on 41% and 55% of the 20 assessments for accuracy and clarity, respectively [42]. Other publications on urology oncology reported that ChatGPT-3.5 and ChatGPT-4 responded in accordance with the guidelines and literature in 80% and 90% of cases, respectively [43] for the tested 30 questions on urology. Our average scores for the 20 tested questions are slightly higher than the published data of ChatGPT-4 responses due to our high-quality knowledge base. Moreover, our 10 generated questions received significantly higher scores: 4.50, 4.50, 4.50, and 4.00 for all criteria, respectively, compared to the 10 real questions. Clinical patient questions are more complicated than the generated ones, resulting in lower scores compared to the generated scores. A report showed that out of 520 treatment suggestions made by ChatGPT-4, oncologists agreed with 322 of them, which is about 62% [18]. Further research on a large number of testing questions is warranted for our retrained LLM.

The novelty of the chatbot lies in incorporating insights from physicians and nurses to address real-world patient questions. We tested 20 questions, including 10 real patient questions and 10 ChatGPT-4-generated questions. We compared the retrained LLM’s performance on real questions with that on the generated ones. Generated questions scored higher in all four criteria, likely due to the straightforwardness of the questions, whereas clinical patients’ queries often present multiple concerns that are more convoluted. The auto-evaluation was employed to compare the LLM’s responses against Epic-based benchmark answers, eliminating human biases while also providing rapid scoring.

This study holds significant implications for supportive care and side-effect management for HN cancer patients by offering intermediate guidance. Delayed access to professional medical information can significantly interfere with a patient’s quality of life; thus, our retrained LLM can provide intermediate and accurate suggestions, potentially alleviating burdens on healthcare providers while also improving patient outcomes. Not only is the great burden on health providers relieved, but the patient’s quality of life can also be potentially increased.

However, limitations need to be noted for this study. The chatbot cannot make referrals or prescription so far. However, it can provide immediate access to medical information and symptom management advice, helping alleviate a patient’s anxiety and improve their quality of life. For instance, for questions 11 and 16, the retrained LLM provides a detailed explanation of the cause of dry mouth and the expected recovery time and tips on managing lymphedema. This study lacked a patient’s evaluation of the retrained LLM in terms of the user experience because the purpose of this study was to validate the performance of a retrained LLM by RadOncs. The number of questions utilized for testing was limited, necessitating further research before any clinical implementations could be considered. Additionally, the RadOncs were not blinded for preference while scoring the LLM answers. The auto-evaluation method cannot address its preference for the answers because GPT-4 does not have the competence to know the correct answer, and a reference answer is compulsory for the evaluation. Another limitation that we consider worth reporting would be that the validation process was carried out by only three qualified oncologists from the same institution, potentially limiting the generalizability of this study. The user-defined knowledge base was collected from the most common questions of HN patients. The knowledge base might not be extensive enough to address all patient questions in clinical settings. A potential harm of the retrained LLM is the collection of protected health information (PHI) [20]. OpenAI’s privacy policy explains that they “may collect personal information included in the input” [44]. We have anonymized the collected questions and answers for both training and testing. In the future clinical settings of our retrained LLM, policies should be established to protect patient privacy. Furthermore, the auto-evaluation method can only be employed to reduce the burden of manual scoring on the RadOncs when there are reference answers to the questions during the validation process of the retrained LLM.

## 5. Conclusions

A custom LLM was investigated and validated for HN cancer patients. It can empower patients with a user-friendly interface for discussing treatment concerns, questions, and experiences. The RadOnc scores of the LLM answers suggest that it has the potential to address real-world clinical patient questions and offer evidence-based information and guidance on survivorship, which is helpful in managing side effects and improving quality of life. The GPT-4 auto-evaluation can evaluate the answers efficiently and rapidly without subjective bias. Further evaluations are needed by real HN patients in terms of the user experience.

## Figures and Tables

**Figure 1 cancers-16-02311-f001:**
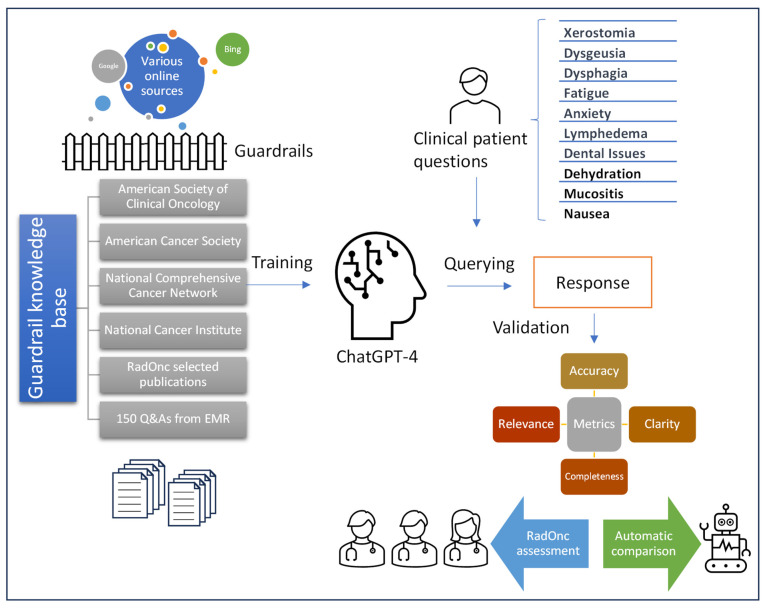
Workflow of a custom-trained chatbot with an external database and the validation process.

**Figure 2 cancers-16-02311-f002:**
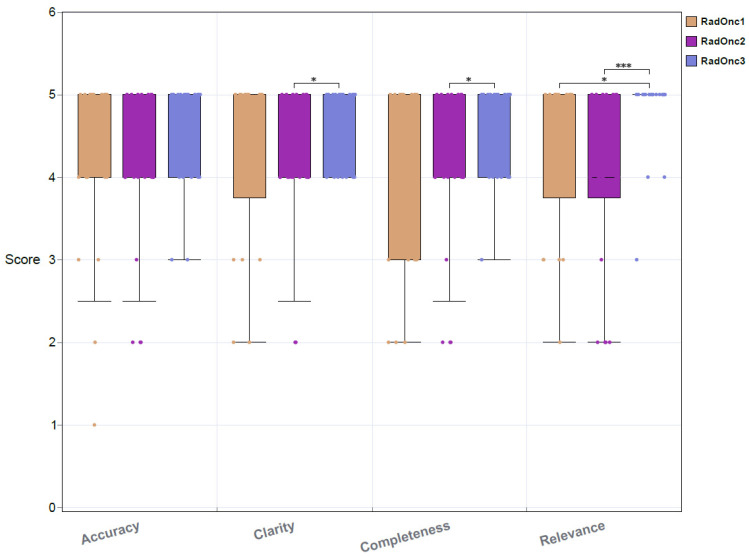
Evaluation of the 20 chatbot answers by three certificated RadOncs (*: *p* < 0.05; ***: *p* < 0.005).

**Figure 3 cancers-16-02311-f003:**
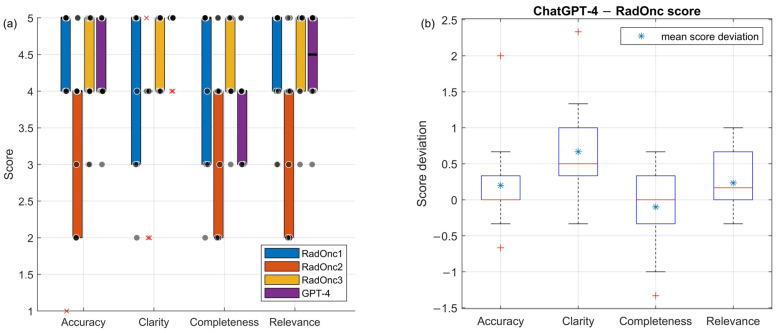
(**a**) Score comparison of the auto-evaluation and different RadOncs on 10 real patient questions (dot sign shows the raw score of each question; multiplication sign is the outlier score in the boxplot); (**b**) score deviation between the auto-evaluation and RadOncs mean score (plus sign indicates the outlier score deviation in the boxplot).

## Data Availability

The original contributions presented in this study are included in the article/Appendix A; further inquiries can be directed to the corresponding author/s.

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
