# Peer review of "Testing and Validation of a Custom Retrained Large Language Model for the Supportive Care of HN Patients with External Knowledge Base"

_cancers, 2024, doi:10.3390/cancers16132311_

Round 1

Reviewer 1 Report

Comments and Suggestions for Authors

Overall a very interesting and novel article. Authors should consider if a better use for the model is as an aide for health professionals, so that simple questions are answered by the chat bot , but those requiring referral or prescription are flagged for human to answer/follow-up. 

Introduction

For completeness, acknowledge literature around Physicians perceptions of the inability of health care chatbots to address the full extent of a patient’s needs (ref:  Palanica et al, Physicians’ Perceptions of Chatbots in Health Care: Cross-Sectional Web-Based Survey, Published online 2019 Apr 5. doi: 10.2196/12887.

Methods

Did the chat bot have ability to look at patient history to inform answers (eg I see you’ve been prescribed xxx)?

2.3 in this section, you state that 20 questions were used. You then state that “separately, 10 additional questions were used”. Can you describe how were these generated? They are not included in results, so should be acknowledged as to why not.

2.3 Can you provide description of what accuracy, clarity, completeness and relevance refer to, perhaps using examples as you do in section 2.4.

Results

Figure 2 add a key or legend describing the significance of the red crosses and the black bars.

The text does not adequately explain the ‘relevance’ result for rad onc 3 (no coloured bar, only red crosses and black bar).

Change description of the supplementary material from ‘additional answers’ to the material ‘includes the 20 questions used and retrained LLM’s responses’. 

Discussion

How can the chat bot save time, when it can’t make referrals or prescribe medications? Many LLM responses recommended going to see a healthcare professional, can you discuss this?

Limitations

The study did not include consumers who could also rate the answers for user experience, suggest this would need to be a future study. 

Potential harm of misinformation as the auto-evaluation can’t know the correct answer without a library of reference answers.

Potential harm includes overlooking patient history/context (see also comment under methods).

The auto-evaluation system, as currently described, can only work with reference to Epic answers. 

Conclusion

Include reference to the need to explore user experience.

Comments on the Quality of English Language

English is clear

Reviewer 2 Report

Comments and Suggestions for Authors

This manuscript describes the development and validation of a retrained LLM targeting symptom management and survivorship care for HN cancer patients.The model showed strong performance as compared to human responses. There are still some areas that needed to be improved in the paper.

1. The author wrote, "We developed a specialized LLM-based model." Did the author's team develop ChuanhuChatGPT, or did the author merely apply ChuanhuChatGPT in the field of HN cancer patients? If the authors only applied ChuanhuChatGPT rather than creating it, it would be more accurate to use terms like "applied," or "utilized.

2. ChuanhuChatGPT is an interface for interacting with several LLMs, rather than an LLM itself. Therefore, the authors need to specify the actual LLM used in the study and the specific hardware environment.

3.The authors' description of the model configuration is mostly conceptual. It is recommended to provide detailed configuration steps using ChuanhuChatGPT. This will help researchers replicate the results.

4. The article format does not fully comply with MDPI guidelines.
